# Towards Circular Economy: Evaluation of Sewage Sludge Biogas Solutions

**Andrey Kiselev** [1,2]**, Elena Magaril** [2] **, Romen Magaril** [3]**, Deborah Panepinto** [4] **, Marco Ravina** [4] **and Maria Chiara Zanetti** [4,*]

[1] Department of Investment Program, Municipal Unitary Enterprise for Water Supply and Sewerage, Tsarskaya str., 4, 620075 Ekaterinburg, Russia; ipkiselyov@mail.ru
[2] Department of Environmental Economics, Ural Federal University, Mira str., 19, 620002 Ekaterinburg, Russia; magaril67@mail.ru
[3] Department of Oil and Gas Processing Technology, Tyumen Industrial University, Volodarskogo str., 38, 625000 Tyumen, Russian; magaril7@yandex.ru
[4] Department of Environment, Land and Infrastructure Engineering, Politecnico di Torino, Corso Duca degli Abruzzi, 24, 10129 Torino, Italy; deborah.panepinto@polito.it (D.P.); marco.ravina@polito.it (M.R.)
[*] Correspondence: mariachiara.zanetti@polito.it; Tel.: +39-0110-907696

**Abstract:** Today it is obvious that the existing linear model of the economy does not correlate with the principles of sustainable development. The circular economy model can replace the current linear economy whilst addressing the issues of environmental deterioration, social equity and long-term economic growth. In the context of effectively implementing circular economy objectives, particular importance should be attributed to wastewater treatment sludge management, due to the possibility of recovering valuable raw materials and using its energy potential. Anaerobic digestion is one of the methods of recovering energy from sewage sludge. The main goal of this study is to make a preliminary evaluation of possible sewage sludge biogas and biomethane solutions using a computation model called MCBioCH$_4$ and compare its results with laboratory tests of sewage sludge fermentation from the northern wastewater treatment plant (WWTP) of Ekaterinburg (Russian Federation). Laboratory experiments were conducted to determine the volume and qualitative composition of biogas produced throughout anaerobic fermentation of raw materials coming from the WWTP. The specific productivity of samples ranged between 308.46 Nm$^3$/t$_{vs}$ and 583.08 Nm$^3$/t$_{vs}$ depending if mesophilic or thermophilic conditions were analyzed, or if the experiment was conducted with or without sludge pre-treatment. Output values from the laboratory were used as input for MCBioCH$_4$ to calculate the flow of biogas or biomethane produced. For the case study of Ekaterinburg two possible energy conversion options were selected: B-H (biogas combustion with cogeneration of electrical and thermal energy) and M-T (biomethane to be used in transports). The results of the energy module showed a net energy content of the biogas between 6575 MWh/year and 7200 MWh/year. Both options yielded a favorable greenhouse gas (GHG) balance, meaning that avoided emissions are higher than produced emissions. The results discussion also showed that, in this case, the B-H option is preferable to the M-T option. The implementation of the biogas/biomethane energy conversion system in Ekaterinburg WWTP necessitates further investigations to clarify the remaining technical and economic aspects

**Keywords:** sewage sludge; biogas evaluation; circular economy; computation modelling

## 1. Introduction

Nowadays the environmental situation is keenly exacerbated, due to the increased anthropogenic load exceeding the ability of the biosphere to support the process of self-regeneration. This crisis

is a consequence of the practice of human consumer's behavior towards the natural environment. Throughout their evolution and diversification, industrial economies have hardly moved beyond one fundamental characteristic established in the early days of industrialization: A linear model of resource consumption that follows a take–make–dispose pattern [1]. Today it becomes obvious that the existing linear model of the economy does not correlate with the principles and goals of sustainable development, creating threats to the existence of future generations [2]. In the last few decades the circular economy has increasingly been advertised as an economic model that can replace the current "linear" economy whilst addressing the issues of environmental deterioration, social equity and long-term economic growth with the explicit suggestion that it can serve as a tool for sustainable development [3]. The concept of circular economy can, in principle, be applied to all kinds of natural resources, including biotic and abiotic materials, water and land [4]. Circular economy is not only an environmental issue, it also affects the way we produce, work, buy and live [5].

In 2015, the European Commission adopted an ambitious Circular Economy Action Plan, which establishes a concrete and ambitious programme of actions, with measures covering the whole cycle: From production and consumption to waste management and the market for secondary raw materials and a revised legislative proposal on waste. These proposed activities will contribute to "closing the loop": Through redesign of production and consumption lifecycles making profit for both the economy and the environment.

Municipal sewage sludge is a specific type of waste that arises in the everyday processes of life, work and leisure activities and during industrial processes [6]. Sewage sludge refers to the residual, semi-solid waste that is originated as a by-product during the process of wastewater treatment [7].

It is important that the results of the activities of the wastewater facilities can be easily accessible by representatives of the professional community and the general public [8]. Annual sewage sludge generation is presented in Table 1 in million tons of dry matter per year (mtDM/year). Sewage sludge is expected to remain a permanent waste problem requiring an appropriate solution.

The most typical technological process of wastewater treatment is presented by the authors in Figure 1. Wastewater passes through a series of treatment steps that use physical, biological and chemical processes to remove nutrients and solids, break down organic materials and destroy pathogens. The rejuvenated water is discharged into the water sources, while solid, semi-solid and liquid waste is retained and concentrated, and sewage sludge is formed.

**Table 1.** Annual sewage sludge generation statistics.

| No | Country | Sewage Sludge, mtDM/year | References |
|----|---------|--------------------------|------------|
| 1 | EU | 13.5 | [9] |
| 2 | Germany | 1.821 | [9] |
| 3 | Poland | 0.568 | [9] |
| 4 | China | 6.25 | [10] |
| 5 | USA | 12.56 | [11] |
| 6 | Russian Federation | 2.5 | [12] |
| 7 | Japan | 2.4 | [13] |

Sewage sludge treatment and further disposal solutions play an important role in the technological process of all wastewater treatment plants (WWTPs). The main goals of these solutions before final disposal include volume decrease of sludge and its organic substance stabilization. Smaller sludge volume reduces the costs of pumping and storage [14].

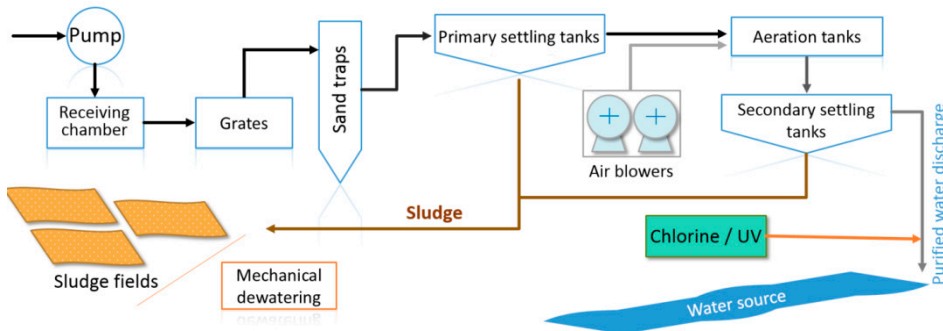

**Figure 1.** Typical technological process of wastewater treatment.

Water content in raw sludge is more than 99% and water removal is the primary solution of weight and volume reduction, while the destruction of the biodegradable part of organic matter is usually implemented through heating during anaerobic digestion, composting, incineration and melting.

Anaerobic digestion and composting involve the decomposition of remnant organic materials. The principal by-products generated from anaerobic digestion are biogases such as methane and carbon dioxide. Composted sludge can be used as a soil conditioner in agriculture and horticulture and returns carbon, nitrogen, phosphorus and essential elements back to the soil [13]. However, the usage of composting in a cold climate is limited; pathogens and heavy metals in composted sludge are also limited.

The European Union has been implementing several policies of urban wastewater and sewage sludge treatment to reduce environmental and health risks and they are considered to be the basic framework for sustainable development and the background for circular economy. The adoption of Sewage Sludge Directive 86/278/EEC in 1986 and the Council Directive 91/271/EEC on urban wastewater treatment in 1991 has led to the increasing quantities of sewage sludge disposal and encouraged the use of sewage sludge in agriculture.

However, there are still many challenges in the area of water and wastewater management in the EU. The suggested actions on better environmental implementation that are related to circular economy include [5]:

1.  Provide further support for local businesses and increase investments in the wastewater treatment sector.
2.  Facilitate development, intensify cooperation and exchange good practices between business units and government entities.
3.  Improve the quality of sewage sludge and its recovery rates.
4.  Optimize energy consumption by sewage systems with the simultaneous production of renewable energy from biogas at the level of wastewater treatment plants.

In the context of effectively implementing circular economy objectives, particular importance should be attributed to sludge management, due to the possibility of recovering valuable raw materials from sewage sludge and the use of its energy potential [15]. The importance of energy recovery in contemporary waste management practices remains assured due to its impact on global waste minimization, resource optimization and alternative energy generation [16].

Anaerobic digestion is one of the methods of generating energy from bio-waste. It involves the transformation of organic matter into biogas in an anoxic environment when acted upon by anaerobic bacteria. Biogas consists of 60–67% methane, 30–33% carbon dioxide, 1–2% hydrogen and 0.5% nitrogen, by volume [17] and can mitigate greenhouse gas (GHG) emissions to the atmosphere.

According to EurObserv'ER Report [9], the production of biogas energy in the EU in 2015 reached 15.6 Mtoe, i.e., 4.2% more than in 2014. While among all the EU countries that produced biogas output figures, almost 77% of Europe's output is concentrated in the hands of three countries—Germany (7.9 Mtoe), the UK (2.3 Mtoe) and Italy (1.9 Mtoe).

Anaerobic digestion modelling and evaluation is of great interest among scientists. These studies are aimed at formation of mathematical equations and models for estimation of biogas yield and the potential of bioenergy to provide information for users (farmers, municipal WWTPs, etc.). The "Anaerobic Digestion Model No. 1" (ADM1) is one of the most popular models, developed by IWA Task Group in 2002. ADM1 includes 32 dynamic state concentration variables, implemented as differential equations [18]. The ADM1 was modified in the study of Zhang et al. [19] by improving the bio-chemical framework and integrating a more detailed physico-chemical framework. The focus is on the design and scale-up of anaerobic digestion units for wastewater treatment and biogas production processes.

Experimental investigations, in contrast to mathematical modelling, explore specific context of biogas yield, e.g., in the study of Adelard et al. [20] two models for estimating methane yield during co-digestion were evaluated. Mirmasoumi et al. [21] explored biomethane productivity at WWTP using three techniques, including pretreatment, digestion temperature rise and co-digestion.

Another group of scientist worked in Life Cycle Assesment (LCA) of sewage sludge, e.g., in the study of Cao et al. [22], a "cradle-to-grave" LCA was conducted to examine the energy and GHG emission footprints of two emerging sludge-to-energy systems. Li et al. [23] conducted LCA alongside economic studies to compare the five anaerobic digestion processes to find out which AD processes are better or best when treating sludge with different organic contents, and give useful information to decision-makers.

Previous studies provide a complete overview of the process of biogas yield evaluation but are quite complex for common users who are interested in applicability of biogas solutions at municipal WWTPs. In addition, there is insufficient information about verification of mathematical models on real sewage sludge biogas plants.

The main goal of the study is to make a preliminary evaluation of possible sewage sludge biogas and biomethane solutions integrating i) laboratory tests of sewage sludge fermentation from northern WWTP of Ekaterinburg (Russian Federation) and ii) simulations using a computation model, called MCBioCH$_4$, for the energetic and environmental analysis. The proposed model, developed at the Department of Environment, Land and Infrastructure Engineering of Turin Polytechnic, Italy, was specifically designed to provide support to the preliminary assessment and comparison of different potential biogas plant configurations and technological solutions. Through this integrated experimental/modelling approach, the objective is the definition of the most efficient and environmentally sustainable sewage sludge conversion scenarios.

The rest of the paper is structured as follows: Section 2 explains the research methodology to identify the study area, characterize modules of the computation model and determine laboratory test conditions. Section 3 presents the results, their interpretation as well as a discussion on them. Finally, Section 4 highlights brief findings.

## 2. Materials and Methods

### 2.1. Study Area

Ekaterinburg is the fourth largest city in the Russian Federation, the administrative center of the Sverdlovsk region and the Ural Federal district, the largest industrial, scientific, educational, commercial and financial center, as well as a transport and logistics hub of the Trans-Siberian Railway. The population of Ekaterinburg is about 1,500,000 citizens.

The centralized sewerage system of Ekaterinburg was built on the basin principle: There are 2 main sewerage zones within the city—northern and southern ones. Wastewater treatment from these zones is carried out at the northern and southern WWTP, respectively. The maximum performance of the northern WWTP is 100,000 m$^3$ per day, while the southern WWTP is 550,000 m$^3$ per day. Mechanical dewatering is implemented both at northern and southern WWTPs. Almost 250 tons of sewage sludge

with a moisture content of 75–78% is formed in Ekaterinburg every day. In other words, more than 90,000 tons every year.

In the last several decades the most typical method of sewage sludge disposal was its placement at specialized landfills, which resulted in overflowed fields with dangerous sediment and offensive odor. Storage of sewage sludge at landfills is accompanied by environmental risks of contamination of surface and underground waters, soils and vegetation. Actually municipal raw sludge is not reused. The existing traditional approach does not meet modern environmental and technical requirements and does not allow the usage of energy and resource potential of waste. Nowadays there is not a single legal landfill for sewage sludge disposal near Ekaterinburg and it is a great challenge for local authorities and municipal enterprises responsible for wastewater treatment.

Since 2007 the Ekaterinburg municipal enterprise for water supply and sanitation has been implementing investment programs for water and wastewater infrastructure development. In 2018 at the northern WWTP in Ekaterinburg, the construction of 2 digesters with volume of 5000 m$^3$ each was finished (Figure 2, authors' photo). Now the company runs test operations and is looking for the best available sewage sludge biogas solutions.

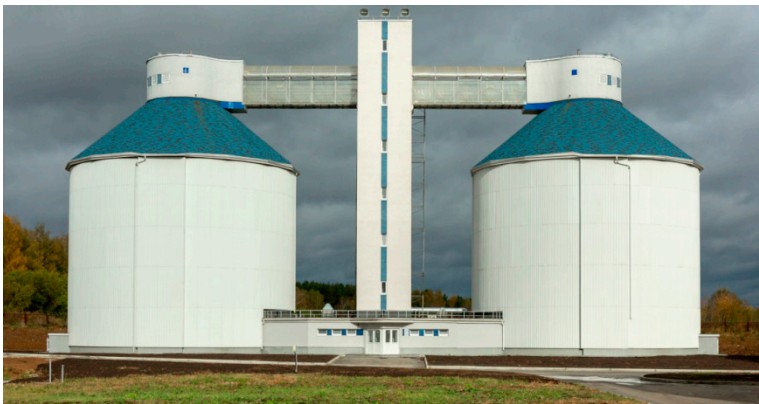

**Figure 2.** Digesters at northern wastewater treatment plant (WWTP) in Ekaterinburg, Russia.

*2.2. Computation Model for Evaluation of Biogas and Biomethane Solutions*

MCBioCH$_4$ (acronym of bio-methane computational model) is a model for the preliminary evaluation of biogas and biomethane solutions. The model focuses on a triple target:

1. Obtaining information about the productivity of biogas/biomethane plants in terms of achievable gas flow rates.
2. Acquiring the plant energy expenditure and subsequently the economically exploitable energy flow shares (electrical and/or thermal energy produced, biomethane being introduced into the natural gas distribution grid or biomethane used as a transport fuel).
3. Accounting for the whole environmental impact of the system on a cradle-to-grave basis, i.e., from substrate production to the end-use of biogas or biomethane as alternative energy sources to fossil fuels.

The design of MCBioCH$_4$ was specifically addressed to provide support to the preliminary assessment and comparison of different potential plant configurations and technological solutions. The code aims at defining the mass, energy and environmental flows referred to the full plant scale. Users are assisted through the implementation of default datasets and an assisted data input.

The computing code has been entirely developed using MATLAB$^®$ software (Mathworks, Natick, MA, USA) and the result is a standalone application fully equipped with graphical user interfaces (GUI). MCBioCH$_4$ was designed with three different modules for the calculation of mass, energy and GHG balance, respectively. Four different possible energy conversion options are implemented:

- Biogas combustion with cogeneration of electrical and thermal energy (option B-H).
- Biogas combustion with generation of electricity only (option B-NH).
- Biomethane to be injected into the national grid at an absolute pressure of 5 atm (option M-G).
- Biomethane to be used in transports, considering a compression and storage system working at 250 bar and consuming electrical power of around 120 kW (option M-T).

If biogas scenarios are selected, the model simulates a combustion in a commercial cogeneration unit (endothermic engine). The recovery of thermal energy can be specified. If biomethane scenarios are selected, the user is allowed to select the upgrading technology, as well as the main features of the upgrading system. The following technologies are implemented: Pressurized water scrubbing (PWS), pressure swing absorption (PSA), chemical absorption with amine solutions (MEA) and membrane permeation (MB). These are considered to be the most common and mature upgrading technologies currently available [24]. Other promising upgrading technologies, such as cryogenic separation (CRY) or those based on carbon mineralization (alkaline with regeneration or bottom ash for biogas upgrading [25]), were not included as they are not commonly diffused at present.

MCBioCH$_4$ is structured with simple and clear dialog boxes in a way that eases the interaction with low-expertise users. As basic starting information, the user is asked to input the daily mass flow of substrates to be inserted into the digester. Other input parameters, specified in the next chapters, can be either provided as default values, or alternatively be specified by the user. The output provided by the model is:

- The detailed mass and energy balance of the system.
- The net mass flow and energy content of the biogas/biomethane stream.
- The greenhouse gas (GHG) balance of the system, including a comparison with an equivalent system powered by traditional (fossil) fuels.

Mass and energy balance of the system may be exported in form of scheme in JPEG format. The complete output of the simulations may be exported in Excel® (Microsoft) (Albuquerque, NM, USA) format. Once inserted, main input information, mass, energy and environmental modules may be run separately and interactively modified. The model also allows the loading of external metadata input files. The structure and main features of the modules are reported in the following.

Compared to other existing evaluation tools, MCBioCH$_4$ presents two main innovations. The first is the calculation of the greenhouse gas flows and balance over the entire bioenergy chain, based on a cradle-to-grave approach. This approach was inspired by life cycle assessment methodologies. Existing models based on LCA (e.g., the BioValueChain, [26]), although being very precise, are usually time-consuming. MCBioCH$_4$ may be considered a simplified LCA approach, with the advantage of a more detailed and faster quantification of the impacts. The second innovation of MCBioCH$_4$ is the detailed characterization of the material entering the digestion process. A large set of existing materials is already implemented in the model, and the possibility of a customized definition is contemplated. The main limitation of this model is that, although it quantifies the digestate production, no further action for digestate management (e.g., estimation of nutrient recovery) is implemented. This aspect will constitute the next step in the development of MCBioCH$_4$.

### 2.2.1. Mass Module

Figures 3 and 4 present logo, entry page and general scheme of the MCBioCH$_4$ developed model.

The mass balance module calculates the flow of biogas or biomethane produced, starting from raw substrates characterization. The parameters that define the biogas yield of each substrate are: Dry matter fraction (DM), volatile solids fraction (VS) and raw biogas yield (biogas volume per mass unit of volatile solids). For the substrates coming from agriculture, the agricultural yield is also needed. Following a detailed bibliographic review, a set of default substrates, representing the most commonly used matter, was implemented in the model (Table 2). Alternatively, customized input materials may

be introduced by the user, as in the case of particular agro-food wastes or municipal solid waste (MSW) organic fractions (OF).

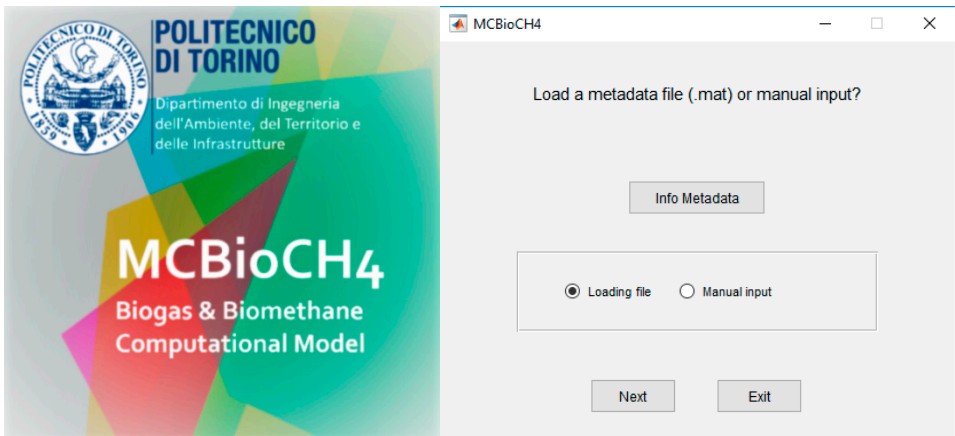

**Figure 3.** Logo and entry page of bio-methane computational model (MCBioCH₄ model).

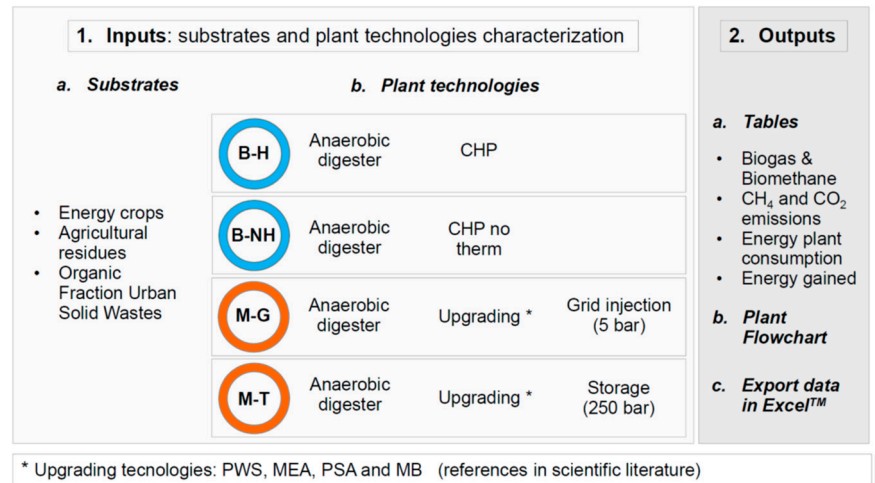

**Figure 4.** General scheme of MCBioCH₄ developed model.

**Table 2.** Substrates implemented in MCBioCH₄ and default yield values.

| Substrate | DM | VS/DM | Biogas Yield (Nm³/t$_{VS}$) | Agricultural Yield (t/ha) | References |
|-----------|------|-------|------------------|-------------------|------------|
| Maize | 0.2200 | 0.9500 | 550 | 60 | [27] |
| Maize silage | 0.3300 | 0.9600 | 700 | 60 | [28] |
| Sorghum | 0.2600 | 0.9600 | 550 | 72 | [29,30] |
| Triticale | 0.3000 | 0.9500 | 625 | 42 | [30] |
| Ryegrass | 0.1800 | 0.8700 | 540 | 55 | [31] |
| Grass | 0.2100 | 0.8700 | 500 | 62 | [32] |
| Cereals | 0.2500 | 0.9500 | 600 | 50 | [33] |
| Beetroot | 0.2300 | 0.9200 | 675 | 100 | [34] |
| Cattle slurry | 0.0800 | 0.7500 | 375 | | [35] |
| Cattle manure | 0.0500 | 0.6800 | 500 | | [35] |
| Swine slurry | 0.2300 | 0.7800 | 290 | | [35] |
| Swine manure | 0.2400 | 0.8300 | 500 | | [35] |
| Poultry manure | 0.6000 | 0.6500 | 375 | | [35] |
| MSW OF | 0.2300 | 0.8700 | 700 | | [36] |

In this module, the digestion process is simulated. The number of digesters is defined according to the inlet mass flow. Users must then specify the temperature of the process (a mesophilic process is set by default) and the fugitive methane emissions from the digesters as a fraction of the net biogas

produced. Fugitive methane emissions from the cogeneration unit (in the case of biogas options) or from the upgrading system (in the case of biomethane options) may also be specified as a fraction of the net biogas produced.

2.2.2. Energy Module

The energy balance module supplies a detailed picture of energy consumption based on different employed technologies and assumptions. Specific energy consumption factors are implemented in the model based on a detailed bibliographic review. Different energy streams of the system are defined following a cradle-to-grave approach, i.e., from substrates production to the final end-use of biogas/biomethane. The selection of such an approach is useful for the definition of the environmental burden of different substrates, performed by the environmental module. In the case of materials coming from agriculture activities, energy consumption of the bioenergy chain is calculated by the use of the specific agricultural yield of the material and a specific energy consumption factor for the selected activity. Energy consumption due to the transport of the substrates to the processing site is calculated by the following parameters: Average distance to be covered (km), transport media capacity (t) and average fuel consumption of the transport media (L/km). In the case of materials coming from waste, a specific energy consumption factor is used to account for waste collection and transport. This factor was defined according to the average capacity of organic solid waste collection media and the average distance expected to be covered from the collection point to the biogas/biomethane site.

The net energy production of the plant, i.e., the conversion of biogas/biomethane to useful energy, is simulated depending on the plant option. If biogas options are selected, the model simulates a combustion in a cogeneration unit (endothermic engine). The size and features of the conversion unit are directly suggested by the model based on a complete set of commercial models proposed by manufacturer Jenbacher. The electrical and thermal efficiency of the engine can be specified by the user. If biomethane options are selected, the useful energy results in the energy content of the methane fraction of the biogas being subtracted from the methane losses from the upgrading process.

If the biogas/biomethane scenario selected includes a production of electricity or heat, the auto-consumption terms are discounted from the gross energy production term. Otherwise, an external energy source is also simulated (electricity grid and/or auxiliary boilers) and the user can specify the conversion efficiency.

Energy auto-consumption (electricity and thermal dispersion) of the biogas section of the system (e.g., to digesters exit) can be calculated following two alternative options: i) They can be defined as a ratio of raw energy output of the system or ii) they can be introduced as an absolute value (MWh/year). If the first option is selected:

- Electricity auto-consumption is calculated by default as 1.3% or 3% of the biogas energy content for an inlet material flow lower or higher than 20,000 t/year, respectively [37]. This value can be customized by the user.
- Thermal energy auto-consumption due to substrates pre-heating and maintenance of the temperature into the digesters is calculated by default as 12.5% or 9.6% of the biogas energy content for an inlet material flow lower or higher than 20,000 t/year, respectively [37]. This value can be customized by the user.
- The amount of thermal energy dispersion to the total heat auto-consumption can also be specified by the user. The default value is set to 20%. This value comes from a publication by Naddeo et al. [38], reporting a range between 13% and 23%, depending on the characteristics of the system.

If biomethane scenarios are selected, the energy consumption of the upgrading process is calculated depending on the upgrading technology, as well as on the main features of the upgrading system. The following technologies are implemented: Pressurized water scrubbing (PWS), pressure swing absorption (PSA), chemical absorption with amine solutions (MEA) and membrane permeation (MB). Consumption is introduced as specific energy (electricity or heat) per volume unit of raw biogas.

The default values reported in Table 3 are proposed. Moreover, if PWS upgrading technology is selected, the energy consumption may also be calculated by introducing the main features of the system. In this case, as reported by Brizio [39] and Ravina and Genon [40], the main contribution to energy consumption is due to the biogas compression and the water pumping. A partial heat recovery from the compressor may also be calculated.

**Table 3.** Default values of specific energy consumption of the biogas upgrading technologies implemented in MCBioCH$_4$.

| Technology | Specific Electricity Consumption (kWh/m$^3$$_{biogas}$) | Specific Heat Consumption (kWh/m$^3$$_{biogas}$) | Heat Recovery from Biogas Compressor (kWh/m$^3$$_{biogas}$) | References |
|---|---|---|---|---|
| PWS | 0.20 | - | 0.11 | [41] |
| MEA | 0.1 | 0.5 | 0.01 | [42,43] |
| PSA | 0.4 | - | 0.03 | [44,45] |
| MB | 0.3 | - | - | [46,47] |

### 2.2.3. Environmental Module

The environmental balance module interacts with mass and energy modules, and provides an estimation of greenhouse gases (GHG) emitted by different plant configurations. Emissions are represented in terms of equivalent $CO_2$ ($CO_2$eq) of the entire complex of activities that directly or indirectly concern the biogas/biomethane plant, based on a cradle-to-grave approach. Substrates introduced into the plant are followed by their cultivation or production up to the final energy conversion. The default emission factors implemented in MCBioCH$_4$ are shown in Table 4.

**Table 4.** Default emission factors implemented in MCBioCH$_4$.

| Phase of the Process | Value | Unit | References |
|---|---|---|---|
| Methane losses (methane GWP) | 28 | kg$_{CO2eq}$/kg$_{CH4}$ | [48] |
| Substrate production (diesel fuel consumption) | Depending on substrate | L/ha | [49,50] |
| Fertilizer production (N/P/K) | 2900/710/460 | g$_{CO2eq}$/kg | [48] |
| N$_2$O emission in agriculture activities | Depending on substrate | kg$_{CO2eq}$/ha | [48,49] |
| Substrates temporary storage | 1.74 | g$_{CO2eq}$/MJ$_{biogas}$ | [51] |
| Substrates transport / handling | 74.1 | g$_{CO2eq}$/MJ$_{diesel\ fuel}$ | [48] |
| Electricity from national grid | 337.1 | g$_{CO2eq}$/kWh$_{el}$ | [52] |
| Natural gas combustion | 206 | g$_{CO2eq}$/kWh$_{th}$ | [48] |
| Fossil fuel mix for transports | 256 | g$_{CO2eq}$/kWh$_{th}$ | [48] |

Specific customizable emission factors are assigned to the different phases of the process. The emission factor of agricultural substrates production and harvesting is calculated as the sum of three components: Fuel consumption in agricultural operations, production and use of fertilizers and N$_2$O emission (direct and indirect). Emission factors associated with fuel consumption are calculated for each substrate based on the specific fuel consumption (L/ha) reported by Cropgen [50] and Astover et al. [49]. Emission factors for fertilizer use are calculated based on average $CO_2$eq emission factors for nitrogen (N), phosphorus (P) and potassium (K) production, considering an average standard N, P and K content. Emission factors for N$_2$O were taken by the IPCC database [48] and Astover et al. [49].

Emissions generated along the biogas/biomethane production process are then compared to the emissions reduction given by the replacement of fossil fuels.

### 2.3. Laboratory Tests of Sewage Sludge Fermentation Process

In order to determine the effectiveness of the process of anaerobic digestion of waste samples obtained from the northern WWTP in Ekaterinburg, the experiments were conducted using laboratory biogas plant (Figure 5, authors' photo). The main goal of laboratory tests was to determine the volume

and qualitative composition of biogas produced throughout anaerobic fermentation of raw materials. Three samples of raw materials were investigated using mesophilic (35 °C) and thermophilic (52 °C) conditions with and without adding of the enzymes of cellulose and lipase: Primary sludge (PS), waste activated sludge (WAS) and a mixture of substrates (PS + WAS) entering the digester.

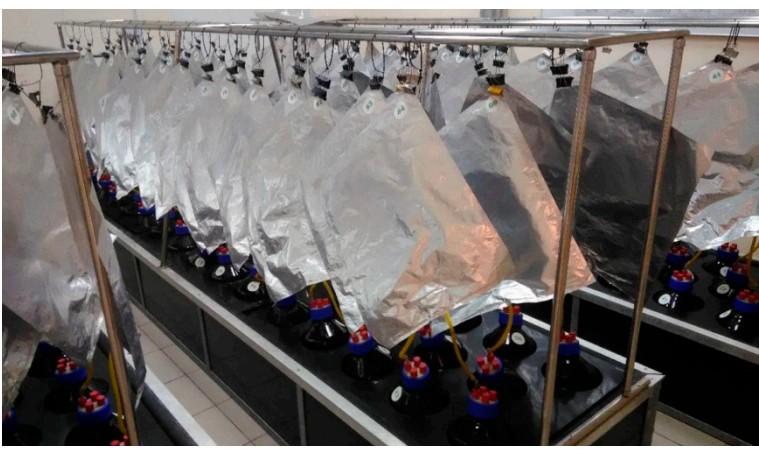

**Figure 5.** Mini-fermenters in laboratory.

The average hydraulic retention time (HRT) was calculated using the equation:

$$\text{HRT} = \frac{V}{\theta} \tag{1}$$

where θ is the amount of feed inside the digesters, and V is the total volume of digesters. Substrate generation at the northern WWTP is 370 t/day, while the total volume of digesters is 10,000 m$^3$. The average HRT is set up for 27 days.

To determine the volume and qualitative composition of biogas obtained in the laboratory from the substrate samples, a special bacterium from the operating biogas plant was added to the mini-fermenters (the same volume for each mini-fermenter) together with each sample. The concentration of enzymes added was calculated as the value needed to load: 200 g of the corresponding enzyme per 1 ton of organic DM of the processed mass.

All experiments were carried out in a triple parallel repetition (i.e., simultaneously, the fermentation process took place in three mini-fermenters for each sample of raw materials). The calculation of the required amount of added mass of raw materials was made according to the method based on the content of organic DM in the samples.

The loaded mini-fermenters were installed in special baths with a constant temperature of 38 °C (±0.5 °C) and 52 °C (±0.5 °C). Before the experiment, anaerobic conditions in mini-fermenters were created using inert gas.

The biogas collection container was connected to the mini-fermenter. The container was periodically removed to measure the volume and quality of biogas produced. The gate valve on the mini fermenter was closed for this time to prevent losses of produced biogas. Each experiment was carried out in a three-fold repetition (simultaneously wandered three mini-fermenters for each type of raw material). The figures obtained for the three containers were averaged.

## 3. Results and Discussions

### 3.1. Environmental Benefits of Biogas and Biomethane Solutions

The main component of biogas (50–75%) is biomethane [53,54], which is a complete analogue of natural gas, and differs only in its origin. On the other hand, only methane is the source of biogas energy. Earlier, it was shown that carbon dioxide emissions could be considered an indicator of the

environmental impact of organic fuels combustion, since they correlate with toxic emissions during the utilization of various types of organic fuels and thus indirectly characterize the dynamics of the pollutant load [55–57]. Thus, the environmental characteristics of methane and other hydrocarbons can be compared on the basis of specific emissions of carbon dioxide for different substances.

The objective characteristic of the fuel in relation to the formation of carbon dioxide in the process of fuel combustion is the ratio of the amount of carbon dioxide produced to the energy obtained. Table 5 shows the results of the calculation by the authors of the lower heating value $Q_l$ and emission of carbon dioxide during the combustion of various hydrocarbons at 300° K. Thermodynamic data were used [58].

**Table 5.** Heat of combustion and specific emission of $CO_2$ during combustion of various hydrocarbons.

| Hydrocarbon | $Q_l$ | | $CO_2$ | | | |
|---|---|---|---|---|---|---|
| | mJ/kg | Relative Units | kg/kg of Hydrocarbon | Relative Units | mg/kJ | Relative Units |
| $CH_4$ | 50.1 | 1.00 | 2.75 | 1.00 | 54.8 | 1.00 |
| $C_5H_{12}$ (pentane) | 45.4 | 0.91 | 3.06 | 1.11 | 67.3 | 1.23 |
| $C_6H_{14}$ (hexane) | 45.2 | 0.90 | 3.07 | 1.12 | 68.0 | 1.24 |
| $C_8H_{18}$ (2,2,4-trimethylpentane (isooctane)) | 44.7 | 0.89 | 3.09 | 1.12 | 69.1 | 1.26 |
| $C_6H_{12}$ (cycloxexane) | 43.9 | 0.88 | 3.14 | 1.14 | 71.5 | 1.30 |
| $C_6H_{12}$ (n-hexene-1) | 44.9 | 0.90 | 3.14 | 1.14 | 70.0 | 1.28 |
| $C_6H_6$ (benzene) | 40.6 | 0.81 | 3.40 | 1.24 | 83.7 | 1.53 |

The calculation results presented in the table show that methane is the most high-calorific hydrocarbon fuel, and at the same time the minimum specific emission of carbon dioxide is generated in the process of its combustion, both per unit of energy produced and per unit mass of fuel, which also indicates the minimum negative environmental impact when using methane, relative to other hydrocarbons, in general.

## 3.2. Biogas Yield

The results of laboratory tests of biogas yield of PS and WAS substrate mixture are presented in Figure 6. Dry matter fraction is 0.033 and VS/DM is 0.7908. According to laboratory tests, the substrate mixture has high VS/DM content and is suitable for anaerobic digestion.

In comparison with Table 1, the results of biogas yield obtained during laboratory tests are below average value. This can be explained by the fact that WAS, which is the part of a substrate that has already passed purification in aerobic conditions, and its VS have low biodegradability. To increase biogas yield with an additional pre-treatment is required

Output values from Figure 6 were used as input values for MCBioCH$_4$ to calculate the flow of biogas or biomethane produced in a potential full-scale plant. Further calculation was made under mesophilic conditions without pre-treatment. For the case study of Ekaterinburg, two alternative energy conversion options were selected: B-H (biogas combustion with cogeneration of electrical and thermal energy) and M-T (biomethane to be used in transports). These options cover two alternative approaches in the design and operating configuration of the plant that reflect on the energy, environmental and economic balance of the whole system. The selection of a solution may depend on several factors that include technical and economic aspects (e.g., location of the plant or economic subsidies introduced by regulations). In principle, onsite biogas combustion in a CHP unit has the advantage of providing useful thermal energy to satisfy the auto-consumption of the plant. On the contrary, these solutions have higher operational costs, and the full utilization of the cogenerated energy is not always possible. Producing biomethane has the advantage of obtaining a versatile energy vector, able to be injected to the national gas grid or used as a fuel transport. The main disadvantage is that an additional external energy source is needed to satisfy the auto-consumption of the plant. For the biomethane (M-T) option, a biogas upgrading with selective membranes (MB) was simulated. Specific electricity consumption of 0.3 kWh/m$^3$$_{biogas}$ (Table 3) was assigned to the biogas upgrading

process. While specific consumption of membrane technologies is higher than other kinds of processes (e.g., PWS), this technology was selected as it is receiving increasing interest thanks to its easiness of installation and low operational costs.

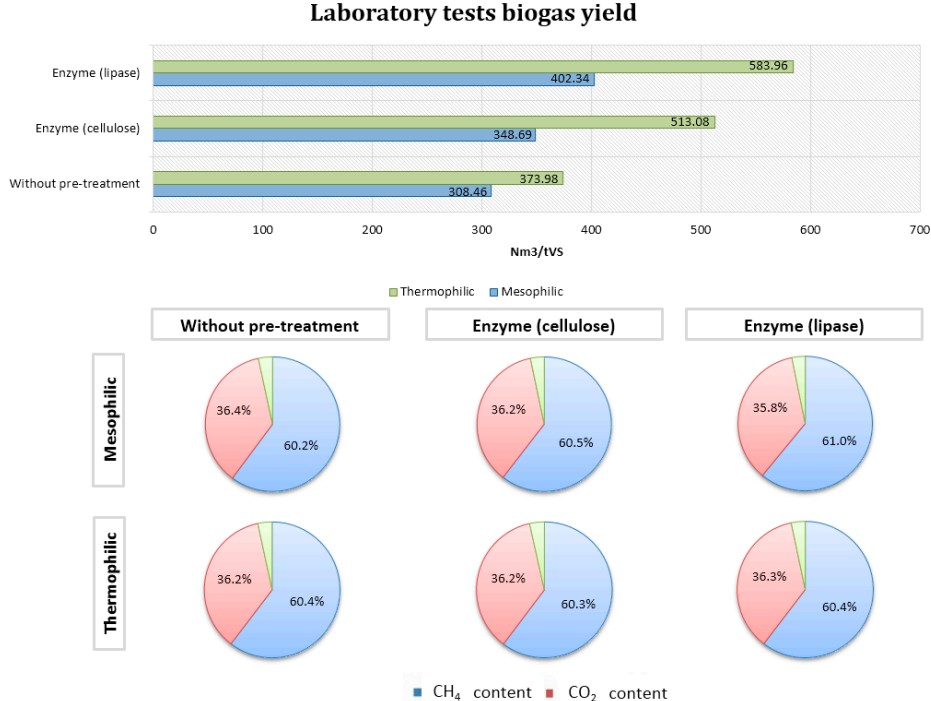

**Figure 6.** Outputs from laboratory tests of biogas yield with hydraulic retention time (HRT) = 27 days.

The principal MCBioCH$_4$ mass balance schemes for B-H and M-T options are shown in Figures 7 and 8, respectively.

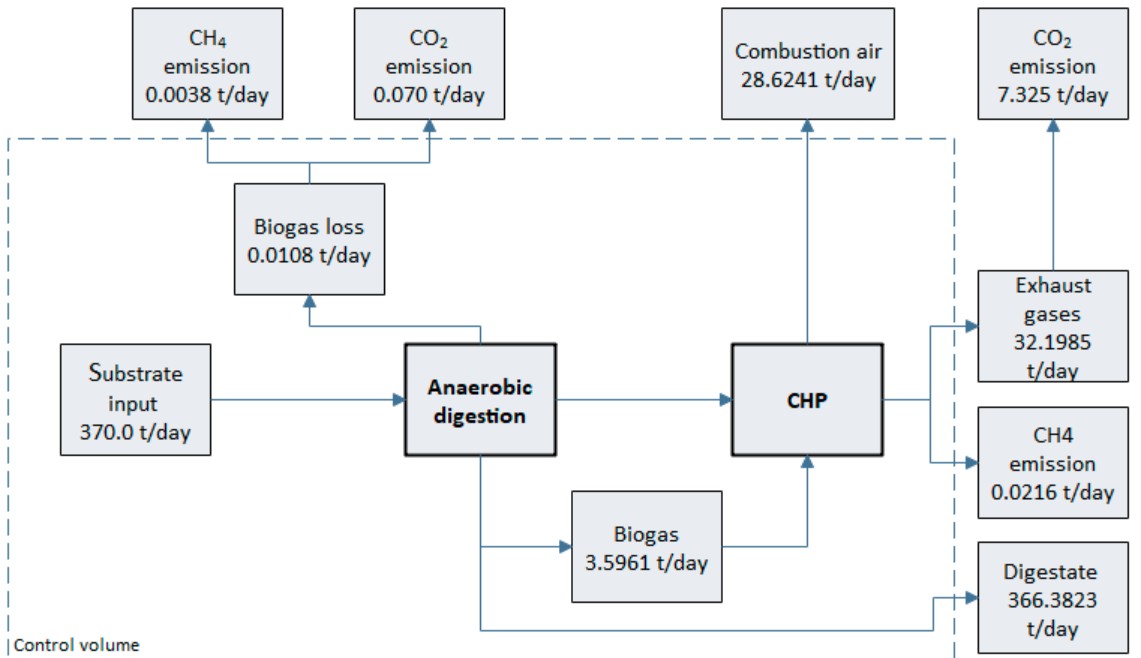

**Figure 7.** Mass balance scheme for B-H option.

Total biogas production for the northern WWTP in Ekaterinburg is 3.5961 t/day (equal to 2969.45 Nm$^3$/day) with consideration of biogas losses from the anaerobic digestion process of 0.0108 t/day. Total CH$_4$ production is 1.2761 t/day (equal to 1787.61 Nm$^3$/day). In the B-H option, part of the methane is lost from the combustion unit (due to fugitive emissions and incomplete combustion). In the M-T option, part of the methane is lost due to the partial inefficiency of the separating membranes. The estimated efficiency of the separation of the membrane system is 98.6%, according to the average values found in literature. The daily flow of digested sludge amounts 366.38 t.

### 3.3. Energy Potential and Environmental Compatibility

The principal MCBioCH$_4$ energy balance schemes for B-H and M-T options are shown in Figures 9 and 10, respectively.

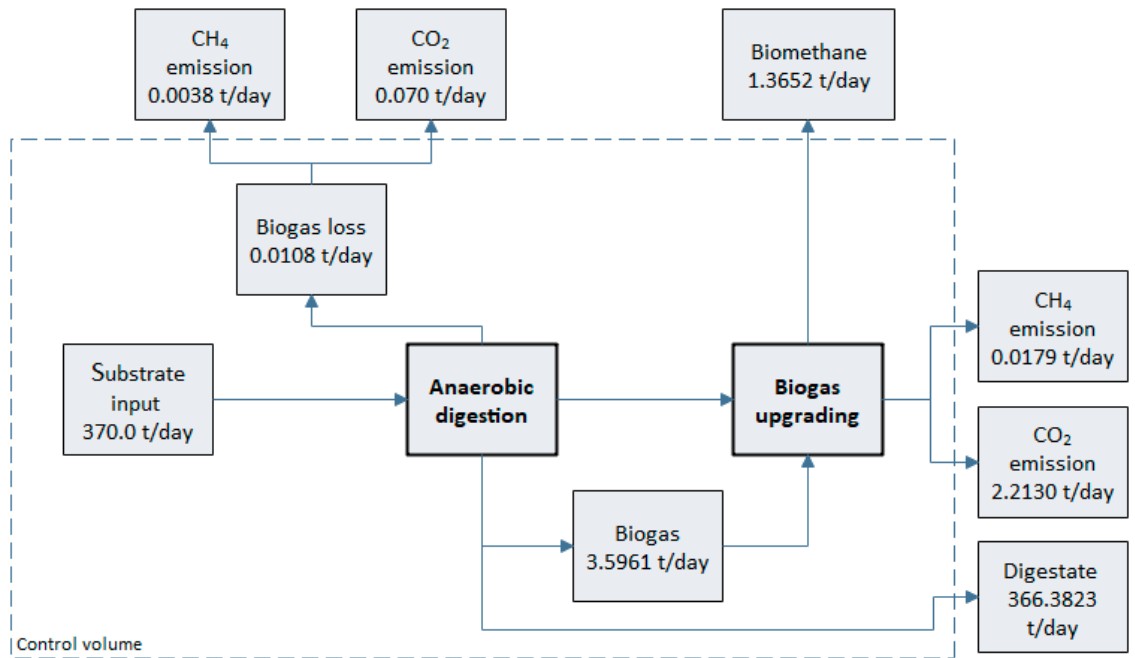

**Figure 8.** Mass balance scheme for M-T option.

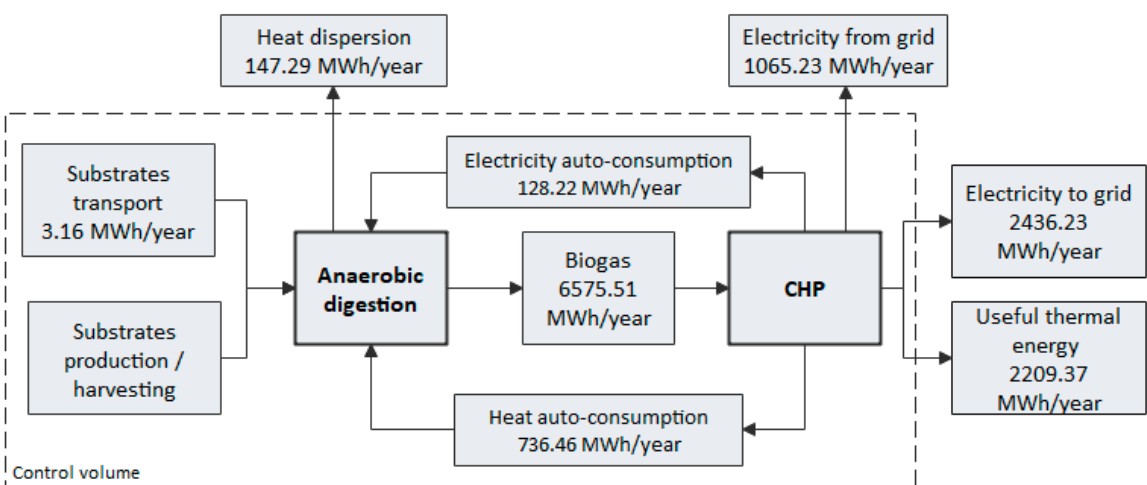

**Figure 9.** Energy balance scheme for the B-H option.

In the B-H option (Figure 9), according to MCBioCH$_4$ calculations, total energy contained in biogas amounted to 6575.51 MWh/year. To estimate energy produced by the combined heat and power

(CHP) unit and fed into the grid, the following characteristics of the CHP unit were used: 0.635 MW of potential electric power with 39.0% of electrical efficiency and 44.8% of heat efficiency. Technological losses of energy for own consumption during the cogeneration process were also considered: Total energy losses amounted to 1065.23 MWh/year. Energy generated by the CHP unit and fed into the grid is estimated at 2436.23 MWh/year for electricity. Useful net heat production is estimated at 2209.37 MWh/year.

In the M-T option (Figure 10), according to MCBioCH$_4$ calculations, total energy contained in biogas amounted to 7200.18 MWh/year. This amount differs to the B-H option due to the higher operating hours per year estimated for the M-T option (8760 h/year, against 8000 h/year of the CHP option). In fact, a full-time operation is usually not possible for the biogas CHP units, as these systems require periodical maintenance. After upgrading biogas to biomethane with 3% of CO$_2$ content, total energy contained in biomethane amounted to 7099.29 MWh/year. A partial heat recovery from the biogas compression stage at the inlet of the upgrading process was also estimated.

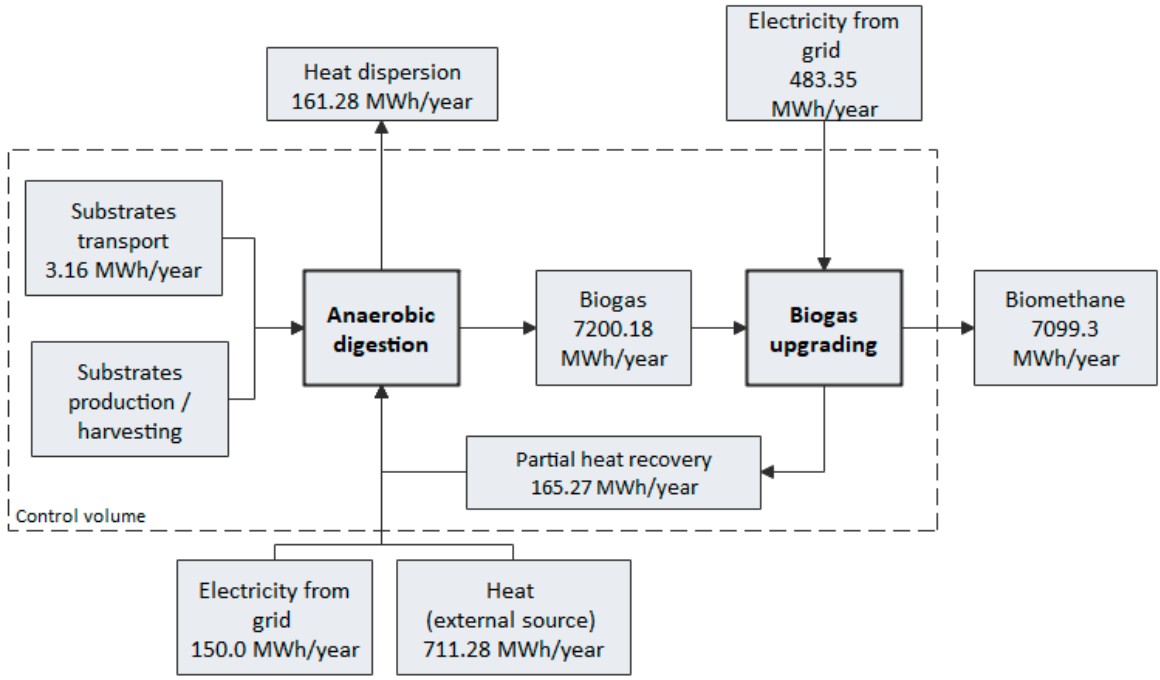

**Figure 10.** Energy balance scheme for the M-T option.

The estimation of the GHG balance provided by the environmental module of MCBioCH$_4$ is presented in Table 6 for the B-H option and in Table 7 for the M-T option.

**Table 6.** Environmental balance for the B-H option.

| Parameter | B-H Option | |
|---|---|---|
| | [tCO$_2$eq/year] [*] | [tCO$_2$eq/t biogas] [*] |
| Emissions from substrate fermentation | 0.00 | 0.00 |
| Emissions from substrate transportation | 0.84 | 0.01 |
| Emissions from substrate temporary storage | 41.19 | 0.03 |
| Emissions from CH$_4$ losses in digester | 35.84 | 0.03 |
| Emissions of unburned CH$_4$ in CHP | 0.61 | 0.01 |
| Emissions avoided due to electricity fed into the grid | −821.25 | −0.68 |
| Emissions avoided due to heat fed into the grid | −455.31 | −0.38 |
| Total | −1198.08 | −0.98 |

[*] Emissions have a positive value, and those avoided have negative value.

**Table 7.** Environmental balance for the M-T option.

| Parameter | M-T option | |
|---|---|---|
| | [tCO$_2$eq/year] [*] | [tCO$_2$eq/t biogas] [*] |
| Emissions from substrate fermentation | 0.00 | 0.00 |
| Emissions from substrate transportation | 0.92 | 0.01 |
| Emissions from substrate temporary storage | 44.47 | 0.03 |
| Emissions from CH$_4$ losses in digester | 39.24 | 0.03 |
| Emission from electricity taken from grid for digesters | 50.57 | 0.04 |
| Emission from heat taken from grid for digesters | 146.58 | 0.11 |
| Emissions from CH$_4$ losses during upgrading to biomethane | 182.58 | 0.14 |
| Emission from electricity taken from grid for upgrading to biomethane | 69.83 | 0.05 |
| Emission from electricity taken from grid for biomethane storage | 93.11 | 0.08 |
| Emissions avoided due to substitution of natural gas for automotive | −1463.04 | −1.11 |
| Total | −835.74 | −0.62 |

[*] Emissions have a positive value, and those avoided have negative value.

Tables 6 and 7 show that both options yield a favorable GHG balance, meaning that avoided emissions are higher than produced emissions. This means that, in general, the energy recovery of WWTP sludge through anaerobic digestion is an environmentally friendly practice and brings a contribution to the development of a circular model of economy. The results also show that the B-H option is preferable to the M-T option under an environmental perspective. Nevertheless, it must be pointed out that the selection of an option depends also on other factors related to the system under analysis. In fact, the B-H option assumes that the thermal energy produced by the cogeneration unit is fully exploited in (or next) the production site. This is the case, as the WWTP and the digestion process require a high amount of process heat. If a total heat recovery could not be possible, the GHG balance would be significantly worse (in previous studies of Ravina and Genon [40], a B-NH option yielded a GHG balance close to zero). Biomethane production has the evident advantage of being a more versatile energy vector, as it is suitable to replace natural gas. Another consideration on the M-T option is that, given the data available, avoided emissions were calculated by replacing only natural gas and not the actual transport fuel mix of the Ekaterinburg/Russian Federation. With this latter emission factor, the GHG balance of the M-T option should improve.

### 4. Conclusions

Anaerobic digestion of sewage sludge provides significant benefits for both companies and society in terms of energy efficiency and environment safety. The proposed approach of the assessment of sewage sludge biogas and biomethane solutions provides an opportunity for WWTPs to choose the most efficient way of 'closing the loop' under a Circular economy context. Applying the results of B-H and M-T options of biogas solution for Ekaterinburg municipal unitary enterprise for water supply and sanitation, cogeneration (B-H) is considered the optimal one. This is due to the following reasons:

- WWTP consumes a large amount of electrical and thermal energy for the processes of wastewater treatment and sewage sludge processing.
- Cogeneration using CHP units is quite a common process and is widely applied all over the world.
- Methane is not popular in the Russian Federation for vehicles and has poor refueling infrastructure; it can be used mainly for corporate transport.
- When upgrading a vehicle to use biomethane, special safety equipment is required (because the methane gas pressure is higher than propane gas) and it is more expensive in comparison with propane equipment. Considering low gasoline price the cost reduction is insignificant.
- The environmental effect of GHG emissions reduction in the B-H option is better than in the M-T one.

The biogas yield can be improved using special pre-treatment techniques to enhance biodegradability of sewage sludge, especially of WAS, or by the construction of additional digester

for a high-calorie substrate (e.g., poultry manure). The decision about the power of a CHP unit for purchase should be made after the feasibility study of the methods mentioned above.

The integrated experimental and modelling approach presented in this study provided a preliminary evaluation of the possible solutions for sewage sludge valorization in terms of energy and matter recovery. The proposed methodology and tools could be, in principle, applied to other similar cases. On the limitations of the methodology presented, it must be pointed out that this study mainly focused on the aspects related to biogas/biomethane production and use, without entering into the details of the nutrient recovery from digestate. In addition, it is expected that other factors not analyzed in the present study (first and foremost economic factors) will contribute to the final selection of the conversion technology. To this end, further investigations to create an additional module of MCBioCH$_4$ for economic evaluations are considered of major importance.

**Author Contributions:** All the authors contributed equally to the present work. Conceptualization A.K., D.P.; methodology—A.K., E.M., M.-C.Z.; software—D.P., M.R., M.-C.Z.; formal analysis—A.K., E.M., R.M., D.P.; investigation—A.K., E.M., D.P., M.R.; writing—original draft preparation—A.K., M.R.; writing—review and editing—E.M., M.-C.Z.; validation—E.M., R.M., D.P., M.-C.Z.; visualization—A.K., M.R.; supervision—E.M., D.P., M.-C.Z.

**Funding:** This research received no external funding.

**Acknowledgments:** This research was supported by Act 211 Government of the Russian Federation, contract No. 02.A03.21.0006.

**Conflicts of Interest:** The authors declare no conflict of interest.

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
