# Peer review of "Towards Circular Economy: Evaluation of Sewage Sludge Biogas Solutions"

_resources, doi:10.3390/resources8020091_

Round 1
Reviewer 1 Report
Please see attached my comments.

Author Response
Dear Mr. / Mrs.
We are truly grateful to you for your deep analysis and appreciation of the paper. You’ll find hereafter answers for your comments.
Here follow some points that need further attention:
Introduction and background sections: Please, be more critical in addressing the research gap. What is the contribution of the paper to the literature? Emphasize these aspects already in the introduction to make paper attractive for readers.
The definition of the objectives has been extended and improved (lines 168-175)
A paragraph with the structure of the paper must be added at the end of the Introductory part.
A paragraph with the structure of the paper has been added at the end of Introduction (lines 176-179)
Conflict of interest must be added before References List.
Conflict of interest has been added to the article before Reference List (line 532)
Limitations of the study must be added to Conclusions
Conclusion has been improved to highlight the significance (and limitations) of this study (lines 516-524)
Line 367 must be deleted
Indicated line has been deleted
All Figures must have mentioned their Sources
Authors have clarified the Figures sources where it was unclear
The Reference list must be reviewed. For example, the publication year must be in bold. While the name of the article should be in italic. Please read the Author’s guidelines and Sustainability template and make the changes accordingly.
Reference list has been reviewed according to the instructions for authors from www.mdpi.com

Reviewer 2 Report
The recovery of nutrients from waste water sludge is indeed an interesting topic and relevant in the context of the tradition to the circular economy. The paper presents a good literature review although very much focused on AD for sludge from MWWTP, without framing it much in the context of the circular economy.
The description of the model is not very clear. After reading the paper I am still confused on whether the authors develop the MCBioCH4 model or this has been adopted from the literature. If the main contribution of the paper is this model, then the model specifications need to be further described. Also, it needs to highlight to what extent this model is better than other existing models to predict nutrient recovery and biogas production. While the biogas production results are described in some detail, there is very little information regarding nutrient recovery through composting or other technologies. There is very little discussion of the technological choices and the benefits and limitations of different technologies, with no references to scale, investment or other contextual factors (e.g. climate).
The discussion and conclusion section there is practically no references to the CE or how the findings from the modelling exercises could be used to make progress towards the CE. Also it is not very clear to what extent the options discussed are based on current or future practices of the waste water treatment plant in Ekateringburg.
Author Response
Dear Mr. / Mrs.
We are truly grateful to you for your deep analysis and appreciation of the paper. You’ll find hereafter answers for your comments.
The recovery of nutrients from waste water sludge is indeed an interesting topic and relevant in the context of the tradition to the circular economy. The paper presents a good literature review although very much focused on AD for sludge from MWWTP, without framing it much in the context of the circular economy.
The description of the model is not very clear. After reading the paper I am still confused on whether the authors develop the MCBioCH4 model or this has been adopted from the literature. If the main contribution of the paper is this model, then the model specifications need to be further described. Also, it needs to highlight to what extent this model is better than other existing models to predict nutrient recovery and biogas production. While the biogas production results are described in some detail, there is very little information regarding nutrient recovery through composting or other technologies. There is very little discussion of the technological choices and the benefits and limitations of different technologies, with no references to scale, investment or other contextual factors (e.g. climate).
The following improvements have been performed:
- Definition of the objectives has been extended and improved (lines 168-175);
- Clear description of the scope, advantage and limitations of MCBioCh4 model has been provided (lines 209-210; 219-222; 255-265)
- Description of the scenarios selected has been improved and provided with technical details (lines 419-436)
Conclusion has been improved to highlight the significance (and limitations) of this study (lines 516-524)
The discussion and conclusion section there is practically no references to the CE or how the findings from the modelling exercises could be used to make progress towards the CE. Also it is not very clear to what extent the options discussed are based on current or future practices of the waste water treatment plant in Ekateringburg.
Conclusion has been improved to make reference to the CE and how the study results could be used to make progress towards the CE (lines 497-499).
Concerning current and future practices of the WWTP in Ekaterinburg: the description of current situation with sewage sludge anaerobic digestion is mentioned in ‘Study area’ (lines 193-205). Study findings revealed the optimal option for biogas solution (lines 499-501). The decision about further implementation of the optimal option for biogas solution should be done after the feasibility study (lines 514-515).
